# Models Used by Nurse Case Managers in Different Autonomous Communities in Spain: A Scoping Review

**DOI:** 10.3390/healthcare12070749

**Published:** 2024-03-29

**Authors:** Paula Villarreal-Granda, Amada Recio-Platero, Yara Martín-Bayo, Carlos Durantez-Fernández, Rosa M. Cárdaba-García, Lucía Pérez-Pérez, Miguel Madrigal, Alba Muñoz-del Caz, Elena Olea, Esther Bahillo Ruiz, Lourdes Jiménez-Navascués, Veronica Velasco-Gonzalez

**Affiliations:** 1Nursing Department, Faculty of Nursing, University of Valladolid, 47005 Valladolid, Spain; paula.vg97@hotmail.com (P.V.-G.); rosamaria.cardaba@uva.es (R.M.C.-G.); lucia.perez@uva.es (L.P.-P.); miguelangel.madrigal@uva.es (M.M.); alba.munozdelcaz@uva.es (A.M.-d.C.); elena.olea@uva.es (E.O.); veronica.velasco.gonzalez@uva.es (V.V.-G.); 2Unidad de Insuficiencia Cardiaca y Terapias Avanzadas, Hospital Clínico Universitario de Valladolid, 47003 Valladolid, Spain; areciop@saludcastillayleon.es; 3Nursing Care Research (GICE), Faculty of Nursing, University of Valladolid, 47005 Valladolid, Spain; esther.bahillo@uva.es (E.B.R.); lourdes.jimenez@uva.es (L.J.-N.); 4Primary Care Management Valladolid West (SACYL), 47012 Valladolid, Spain; 5University Clinical Hospital of Valladolid, 47003 Valladolid, Spain; 6Instituto de Biomedicina y Genética Molecular (IBGM), Consejo Superior de Investigaciones Científicas, Universidad de Valladolid (UVa-CSIC), 47005 Valladolid, Spain; 7Nursing Department, Faculty of Health Sciences, University of Valladolid, 42004 Soria, Spain

**Keywords:** advanced practice nursing, case management, national health system, nursing care, scoping review, Spain

## Abstract

(1) Background: The role of the nurse case manager is unknown to the population. The main objective is to analyze the existing differences within the national territory in order to make known the situation in Spain with a view to the recognition of its functions and the creation of the professional profile in an equal manner. (2) Methods: A scoping review was conducted in order to achieve the main aim. Selected articles were subjected to a critical reading, and the levels of evidence and grades of recommendation of the Joanna Briggs Institute were verified. The search field was limited to the last ten years. (3) Results: Case management models are heterogeneous in different autonomous communities in Spain. Case nurse management is qualified for high-complexity patients, follow up on chronic patients, and coordinate health assistance. (4) Conclusions: It concludes that nursing’s role is crucial in the field of case management, being required in the follow-up of chronic patients of high complexity. Despite the proven health benefits, efficacy, and efficiency of case management, there are many heterogeneous models that coexist in Spain. This involves a restriction in the development of a nursing career because of the lack of a definition of its functions and competences.

## 1. Introduction

Case management (CM) is used to promote better coordination of health services and allow greater access to all populations; it optimizes available resources, improves communication between health care professionals, patients, and their families, and improves the quality of care [1,2]. CM is known internationally as a strategy to reduce health costs, and there is now strong evidence that the implementation of CM can reduce hospital use and improve patients’ quality of life [3,4,5,6,7]. CM began in the USA in the 1960s as a new service delivery model. Initially, the model was used for public health actions [8]. Subsequently, changes in the care of psychiatric patients led to the need for a systems agent in the 1970s in this country. Already in the 1980s, CM was extended to complex patients. The Health Maintenance Organization in the United States and the National Health Service in the United Kingdom were the driving forces behind the advancement of CM-focused nursing [9,10].

The case management model is concerned with the short-, medium-, and long-term care, community involvement, and support of a person with different health problems in a variety of settings. However, despite all of the above, some authors refer to the fact that defining case management has often been more challenging than defining other professional nursing roles [11]. An essential professional to consider is the Advanced Practice Nurse (APN), defined by the International Council of Nursing (ICN) [12] as “a specialist nurse who has acquired the expert knowledge base, complex decision-making skills and clinical competencies necessary for extended professional practice whose characteristics are given by the context or country in which the nurse is accredited to practice. An intermediate university degree is recommended as an entry level”. Therefore, a nurse case manager (NCM) performs her/his daily practice with APN competencies [13].

The nurse plays a key role in case management. Commonly known as NCM, is responsible for case identification, multidimensional assessment, care delivery, and evaluation of the health outcomes of the patient and their family members [14]. The scope of action of the nurse case manager is very broad; including primary care, specialized care, and home care [1,14,15]. For a long time, there has been confusion about the role of NCMs internationally due to constantly changing models of CM, no consensus on its definition, and little theoretical underpinning. Although CM programs have been widely introduced in healthcare systems internationally, the roles of CM are not always clearly defined, and there is a lack of standardization of roles [16,17]. As for the definition of CM, there is no international agreement. The American Society of Case Management defines it as “*a collaborative process of assessing, planning, implementing, coordinating, monitoring and evaluating the options and services necessary to meet an individual’s health needs, articulating the communication and resources available to promote quality and cost-effective outcomes*” [18], but the Ontario Society of Case Managers defines CM as “*the collaborative process of providing health and support services through the effective and efficient use of resources. Case management strives to achieve real and reasonable goals in a complex health and social environment*” [19]. In Spain, the definition seems to be more homogeneous, but on the other hand, the Spanish health services that have this type of nurse do not have unified functions and are subject to different regulations. For example, Andalusia frames it within a Family Support Act and the Balearic Islands within the Community Care Strategy of the Spanish Ministry of Health [20,21].

In Spain, CM is an advanced nursing care practice and a basic strategy for chronic care, focusing on the care of highly complex patients. Its objectives are aimed at improving the quality of care and reducing costs, guaranteeing continuity of care, and facilitating accessibility and integration of multiple services. However, the lack of international consensus on the definition and competencies of APN, together with the different models of case management coexisting in Spain, represent a considerable limitation in the consolidation of this nursing role [22,23]. Law 14/1986, of 25 April [24], of the General Health Act establishes universality of coverage as one of the basic characteristics of the Spanish system by determining that public health care will be extended to the entire population. It also establishes equity as a general principle of the National Health System (NHS), understood as the guarantee that access and healthcare services will be provided under conditions of effective equality. The NHS in Spain is characterized by its extensive decentralization, which culminated definitively in 2002, when all the autonomous communities (a total of 17) were granted the healthcare competencies provided for in our legal system, with each one having its own health system [25]. This NHS, whose decentralization facilitates better adaptation to the health needs of patients and users, also requires the development of cohesion and coordination actions to ensure the appropriate application of common strategies and measures throughout the territory [25]. Despite this, the NCM is not equally integrated throughout Spain, with confusion as to its roles and a lack of consensus on its functions, as well as differences in the recognition and integration of this professional profile within the health system of each autonomous region [21]. The first implementation of this figure in Spain was in the Canary Islands and Catalonia in 1994. At present, seven Spanish autonomous communities have NCMs. In most cases, the NCM is a nurse exclusively for these functions, but in Madrid, for example, this is not the case. In this context, the CM is the competence of nurses with a primary care profile as far as chronic care is concerned [26]. Therefore, the objective of this review is to analyze the existing differences within the national territory of this professional profile in order to make known the situation of the NCM in Spain with a view to the recognition of its functions and the creation of the professional profile in an equal manner throughout the Spanish territory.

## 2. Materials and Methods

### 2.1. Design

A scoping review was conducted following the methodological framework proposed by Arksey and O’Malley [27] and Levac et al. [28]. The review process was divided into five stages or phases: (I) identifying the research question; (II) identifying relevant studies; (III) study selection; (IV) data extraction; and (V) summarizing and reporting the results. In addition, the recommendations of the “Preferred Reporting Items for Systematic Reviews and Meta-analysis extension for Scoping Review” (PRISMA-ScR) were followed [29], and the levels of evidence and grades of recommendation established by the Joanna Briggs Institute (JBI) were verified [30].

### 2.2. Phase I: Identifying the Research Question

With this scoping review, the objective is to explore the range of available literature on a specific topic. This method was deemed suitable for addressing the following research question: What are the characteristics of the models used by nurse case managers in the different Spanish autonomous communities in relation to their professional roles?

### 2.3. Phase II: Identifying Relevant Studies

The literature search was carried out in different databases: Medline, Scielo, and Web of Science (WOS), completing the research in the virtual library of Cochrane Library and the electronic journals of ScienceDirect, and incorporating specific documentation of the different web portals of the different health services of the Spanish territory and official documentation of the Ministry of Health website. This search was carried out by means of Descriptors in Health Sciences (DeCS) and Medical Subject Headings (MeSH), in each of the aforementioned sources, together with natural language. These descriptors mentioned were “Enfermería”, “Enfermero/a”, “Enfermería de Práctica Avanzada”, and “Gestión de Casos” in Spanish, as well as “Nursing”, “Nurse”, “Advanced Practice Nurse”, and “Case management” in English, together with the Boolean operators AND and OR.

The search strings used were as follows:Nurse OR Nursing AND Case Management;Nurse OR Nursing AND Advanced Practice Nursing;Enfermero/a OR Enfermería AND Gestión de Casos;Enfermero/a OR Enfermería AND Enfermería de Práctica Avanzada.

### 2.4. Phase III: Selection of Studies

The bibliographic search was carried out during the months of November and December 2023, initially limiting the search field to the last five years. Due to the limited availability of written documentation related to the research question within Spanish territory and for the bibliography to be updated, the decision was made to expand the search to include documents published in the last ten years, when the role of the NCM begins to be implemented in Spain and is included in strategies for addressing chronicity.

Regarding inclusion criteria, we opted for publications that addressed case management by nursing staff within the Spanish territory, were written in either English or Spanish, and were published in the last ten years. The documents incorporated into the scoping review primarily comprised systematic and bibliographic reviews, comparative analysis, descriptive studies, expert opinions, and chronic care programs. Documentation pertaining to case management beyond the national context and/or published over the last 10 years was excluded from this review, except for specific chronicity management strategies, which were included if published before 2013 due to the lack of recent updates.

### 2.5. Phase IV: Data Extraction

For data extraction, a table was employed to consolidate key information from the selected papers, facilitating the identification of main findings relevant to the research question. The table encompassed details such as author and year of publication, sample size, primary outcomes, study design, level of evidence, and grades of recommendation as established by the Joanna Briggs Institute (JBI) [30].

### 2.6. Phase V: Compilation, Summary and Communication of Results

Initially, a descriptive analysis (frequencies and percentages) was conducted based on the design of each of the selected documents. Furthermore, these documents were analyzed in accordance with the review’s objectives, delineating the sample characteristics and highlighting the main results pertaining to case management.

## 3. Results

Of the 135 documents found, 27 were included after full text analysis (Figure 1). Of these, 3.7% (n = 1) were systematic reviews, 11.1% (n = 3) were literature reviews, 7.4% (n = 2) were management models, 3.7% (n = 1) were analysis papers, 51.85% (n = 14) represented strategies for approaching chronicity, 14.8% (n = 4) were opinion articles, and the remaining 7.4% (n = 2) were descriptive studies.

All the autonomous communities in Spain have strategies for dealing with chronicity, except for two, which do not include the case management model as an instrument for dealing with it. CM is a strategy used internationally in the approach to chronicity, allowing continuity of care and reducing the number of hospital admissions of the most complex chronic patients, which leads to a reduction in healthcare costs [31,32]. In addition, the nurse is the main professional designated to carry out and lead case management, which is why new health management models are needed to include this professional role in daily practice [33,34].

Some of the co-existing CM models in Spain have evaluation processes that allow their effectiveness to be assessed; to date, they have been consolidated in three autonomous communities (ACs) [35,36].

### 3.1. Nurse Case Manager

Only two autonomous communities (11.8% of the total) make no reference to leadership by nursing professionals in case management. Although one of them proposes the creation of new professional roles, none of them refers to the role of nurses as a key figure in the management models [35,36]. In contrast, Andalusia, Valencia, Catalonia, and the Basque Country are the communities where the nurse case manager role is more established. Its functions are well-developed, and there is evidence of the implementation and operation of this professional role [37,38,39] (Figure 2).

In Table 1, the main results of the articles on case management in nursing are described. Most of them demonstrate the benefits of case management in healthcare practice, often focused on chronic care. This case management is oriented towards identifying and caring for patients with high dependency and complexity, addressing the needs of patients with complex conditions stemming from chronicity, multi-pathology, frailty, and aging [22,40]. Case management as an advanced practice has become a basic strategy in the care of complex chronicities in Spain, guaranteeing multi-professional, coordinated, and evidence-based care and promoting coordinated actions and strategies for the same territory and population. These professionals have very heterogeneous functions, and they should be focused on guaranteeing continuity and education for self-care [31,32,36].

### 3.2. Implementation of a New Nursing Role

Three ACs, accounting for 17.6% of the total, do not consider NCM as a new category within the health system; two of them (11.8%) refer to the assumption of this professional role by primary care (PC) nurses, while the other (5.8%) only mentions the performance of CM by nurses [43,44].

The remaining communities (82.4%) propose a new nurse figure for CM: two (11.8%) propose a nurse case manager attached to PC, while another eight (47%) include two new CM roles: PC CM and specialized care (SC), also known as a liaison nurse who guarantees continuity of care [21,22,37,38,39,43,44,45,46,47,48,49,50,51,52,53].

On the other hand, specific functions for NCM are only described in 8 ACs (47%); concurrent with the creation of a new professional profile for addressing chronic patients and promoting patient autonomy and self-care (Figure 2).

Table 2 shows the main results of chronic disease management strategies by nurses in the different regions of Spain. The Ministry of Health’s strategy for addressing chronic conditions dates back to 2012, aiming to standardize criteria regarding key aspects of care for chronic patients, serving as a guide for the rest of the autonomous communities [35]. On the other hand, one of the most current strategies is that of the Andalusian Health Service, which aims to integrate services based on patient needs, plan care, and coordinate different levels of care [37,49]. Cantabria, C. Valenciana, Madrid, and La Rioja propose the creation of a new nursing professional profile for the care of chronic patients. What they all agree on is the need to provide quality care and the complexity involved, promoting interdisciplinary teamwork, and guaranteeing continuity of care with the maximum participation of the patient and their environment [35,39,46,51,53].

### 3.3. Evidence of Case Management

There is evidence that eight autonomous regions (47%) have implemented case management. As is known, the Autonomous Community of the Canary Islands was a pioneer in promoting CM in Spain, although it did not standardize the figure of the NCM until 2002, the year in which a second autonomous community implemented this figure [22,40,48].

Several pilot projects have been developed in three communities (17.6%). After analyzing them, two NCM figures have been consolidated in one of them: the hospital liaison nurse manager and the advanced competency nurse manager (Table 1).

In the capital of Spain and in an island territory, the role of hospital liaison nurse has been incorporated in several hospitals. Another community, after several years of working with liaison nurses, abolished their figure in 2013, considering that PC nurses can carry out this type of function [22].

Of all the existing autonomous regions, only five of them (representing 29.4%) present measurable data on the functionality of NCM, highlighting positive variables related to the perceived quality of life of patients and their caregivers, the degree of satisfaction with the care received, the overload of their caregivers, the degree of functional dependence of the patient, and the consumption of health resources [22,40,41].

Only three (17.6%) define the functions of the NCM in a protocolized manner, with only the functions of one of these ACs being up-to-date (Figure 2).

## 4. Discussion

Based on the findings and stated objective, the results of this literature review are discussed. The implementation in Spain of the NCM figure is not regulated by regulations that provide stability to a care model that seems proven to be adequate, and its heterogeneity is more than evident [21,35].

This diversity has repercussions on the performance of the health professions; in the case of nursing, its competencies and functions are not defined or protocolized under a legal framework that endorses and recognizes the importance of the nursing profession in the monitoring of chronic diseases. This hinders the advancement of the profession and the recognition of its functions, pushing case management by nurses into a secondary role despite numerous studies supporting the effectiveness of activities performed by nurse case managers as vital contributors to care coordination for chronic patients. This is evidenced by patient satisfaction, continuity of care, and reductions in emergency visits and hospital admissions, among other indicators [26,31,54].

Another crucial aspect to consider in case management is cost reduction. A study conducted in the Valencian community illustrates the potential savings associated with the involvement of nurse case managers in the discharge management of patients admitted to chronic hospitals. The estimate suggests that between 4.4% and 19.4% of additional patients could be attended to without incurring extra costs, although further studies are needed to confirm this [42,55]. Other research focused on the analysis of case management in the Canary Islands shows improved clinical outcomes and greater efficiency in the use of resources while promoting coordination with social workers [31]. In the autonomous community of Andalusia, the study ENMAD demonstrates the effectiveness of case management models in home care, highlighting the improvement of the autonomy of immobilized patients, reducing care costs, improving the therapeutic plan, and coordinating with social workers and physiotherapists, thus forming a multidisciplinary team [56].

If the situation of NCM is extrapolated to the international level, a systematic review of this professional profile carried out in the United States of America also mentions the lack of consensus and clarity regarding the definition of the role, as well as the lack of a theoretical framework, despite the fact that case management is a practice that has been carried out for decades in the country. Another significant point in this research is the scarcity of documentation on the specific role of NCM and the resulting confusion among nurses. Despite this, reported benefits include improvements in continuity and transition of care, patient-centered care, and a reduction in healthcare costs, mirroring the situation in Spain [57]. In the European context, there is evidence that describes the professional profile of nurse case managers in numerous articles. However, there is no unanimity regarding the degree of specialization, the definition of its functions, or the legal framework, which makes it difficult to compare the status of this professional profile in different European countries. Evidence has been found on the Advanced Practice Nurse, which, as mentioned above, encompasses the case management profile, with disparities between European countries in terms of the implementation of the APN role. The Netherlands is one of the countries with the highest percentage of APNs per 100,000 inhabitants and one of the countries that is committed to the case management model in different areas such as palliative care and mental health, while most European countries are in the early stages of implementing these nursing roles [58,59,60]. A study carried out by means of a review of reports from European countries determined that a certain degree of specialization is required to be able to carry out the functions of a NCM [61]. In Spain, there is no nursing specialization in this area, but there is a tendency to consider these nurses as advanced practice nurses, even though there is no jurisprudence to support this [26]. Outside of Europe, there are emerging NCM models in Latin America. Colombia has nursing practices that could be considered advanced but are not recognized as such. Some nurses perform liaison or CM functions, but the country’s legislation does not yet allow them to be formally recognized. Chile has a situation like that of Colombia [62].

The debate over whether a specialist nurse or an advanced practice nurse is more appropriate remains open. In general, the literature shows an increasing commitment to the APN. In Europe, the first APN function emerged in the United Kingdom. The Netherlands has a long history in this field, as does Finland. Iceland is committed to hospital specialization, but changes are also being considered in community health. In Asia, the APN concept is emerging in several countries, such as Japan, Hong Kong, South Korea, the Republic of China, Taiwan, and Singapore. Universities in Pakistan have shown interest in initiating the development of advanced practice, but efforts have been diverted towards training nursing managers and teachers. The Nursing Council of New Zealand accredits clinical master’s programs for advanced practice nurses. Australia already has health results that have been achieved since this practice was introduced and promoted [63].

This review has both strengths and limitations. The primary and most significant limitation is the inability to generalize the situation of nurse case managers at an international level, as the review is based on the existing professional nursing profile in Spain. Another important limitation to consider is the absence of updated strategies for addressing chronic conditions. At the national level, the latest available guide dates back to 2012. It is worth noting that in 2021, an evaluation report and priority lines of action were formulated for the same guide [35]. Regarding the strengths, while this professional profile is not uniformly defined or standardized across Spain, the review provides a comprehensive and precise overview of the NCM situation in each of the Spanish autonomous communities, offering a comprehensive understanding of this professional profile across the national territory. Although more research is needed on this professional profile, the perspective provided on case management in Spain can lay the foundations for unifying and protocolizing the competencies and functions of the NCM by means of a legal framework that supports them, thus guaranteeing the consolidation of the NCM profile in an equitable manner throughout the territory and thus allowing for a great advance in the nursing profession.

## 5. Conclusions

In conclusion, nurses play an indispensable role in the field of case management, being a fundamental figure in the continuous care of highly complex chronic patients. The implementation of case management models can have a positive impact by decreasing hospitalizations and readmissions, delaying nursing home admissions, improving patient satisfaction with the health care system, and alleviating the burden on non-professional caregivers. Although case management has been shown to benefit the health care system and patients with chronic diseases, its implementation in Spain is not regulated by legislation to give stability to this model of care. Moreover, there is no legal framework in Spain to support NCMs, which relegates nursing to a secondary role despite the fact that scientific evidence has shown that its care of the chronically ill is essential.

## Figures and Tables

**Figure 1 healthcare-12-00749-f001:**
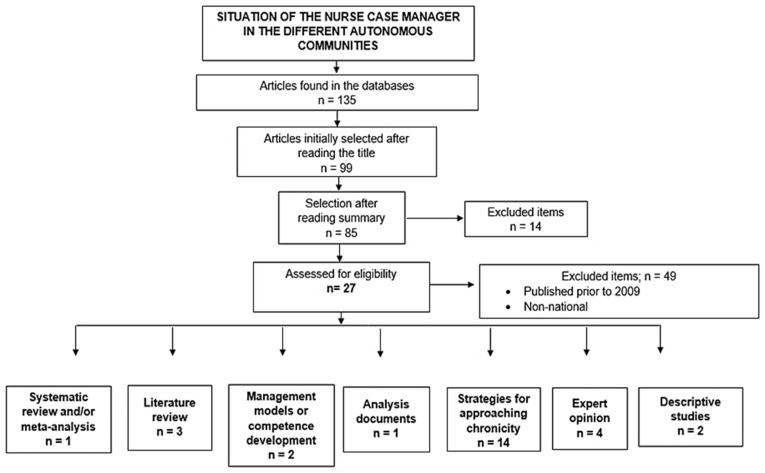
Flowchart depicting the article selection process.

**Figure 2 healthcare-12-00749-f002:**
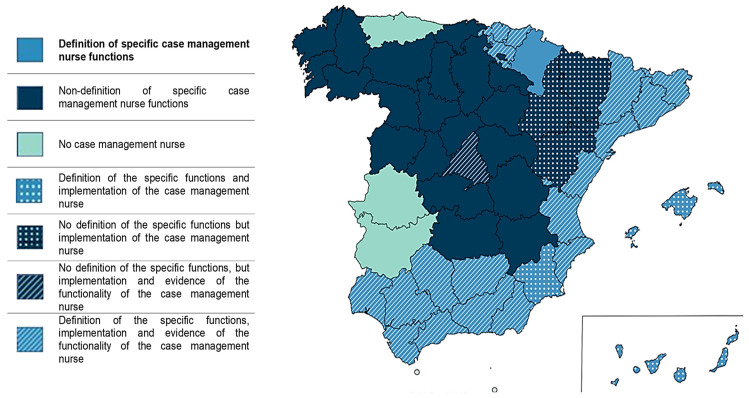
Nursing functions, implementation, and functionality of case management in Spain.

**Table 1 healthcare-12-00749-t001:** Results of the articles on case management in nursing in the different autonomous communities of Spain.

Author/Year	Sample	Main Results	Design	JBI *	JBI **
Martínez et al., 2013 [41]	NCMs	Discharge planning, with a comprehensive individualized assessment, reduces readmissions and length of stay in hospital in people over 65 years of age.	DS	4	B
Comellas-Oliva, 2016 [40]	APNs	The case management model is oriented towards the detection and care of patients with high dependency and/or care complexity, integrating the figure of the nurse case manager.	EO	5	A
Sánchez, 2014 [22]	APNs	Advanced practice nurses and the incorporation of new roles (NCM) respond to the demand for care from patients with complex situations arising from chronicity, multi-pathology, frailty, and aging.	LR	4	A
Enfermería en Desarrollo. 2014 [42]	CPs	An integrated care model for complex cases in the Valencian community has reduced patient visits to the emergency department by 77% and hospital admissions by 70%, increasing the satisfaction of healthcare professionals.	EO	5	A
Romero, 2014 [32]	CP and NCMs	The functions of these professionals are very heterogeneous; they are responsible for the care plan, coordinate the work of other professionals, and participate in the distribution of care and social and health care resources.	EO	5	A
Miguélez-Chamorro et al., 2019 [31]	CPs	Case management, as an advanced practice, has become a basic strategy in the care of complex chronicity in Spain, guaranteeing multi-professional, coordinated, and evidence-based care, promoting coordinated actions and strategies for the same territory and population. NCM ensures professionalized, comprehensive, and continuous care for the most vulnerable people.	LR	4	A
Morales-Asencio, 2014 [36]	CPs	Andalusia, Catalonia, and the Basque Country are the ACs in which NCM is most consolidated. There is no global evidence available regarding its effectiveness. The role and responsibilities of the case manager should be defined and focused on guaranteeing continuity and education for self-care.	LR	4	A
Mármol et al., 2018 [26]	CPs	NCM is a key figure in the care of complex chronic patients. The activities carried out by this professional are more effective and efficient in the approach to patients with chronic pathologies than those carried out following the traditional method.	SR	2	A
López. et al., 2019 [21]	NCMs	The implementation of NCM varies from one autonomous region to another, with great heterogeneity. All the autonomous regions except Asturias, Extremadura, Ceuta, and Melilla consider nursing to be in charge of case management. In turn, Castilla La Mancha, Castilla y León, and Galicia do not consider the inclusion of NCM a new category within the health system. The rest of the autonomous communities present NCM as a new professional figure and role and implement it in different ways.	CA	3	A
Lapeña-Moñux et al., 2017 [33]	NCM	Consolidation of nursing leadership in case management, with the need to develop nursing competencies, making their role as professionals in the care of complex chronic patients more visible.	DS	4	B
San Martín-Rodríguez, 2016 [34]	APN	Spain’s healthcare systems have made progress in developing new organizational models to meet the changing needs of today’s population, reducing healthcare spending, and increasing the quality of management.	EO	5	B

* level of evidence; ** grade of recommendation; JBI: Joanna Briggs Institute; CPs: chronic patients; NCMs: nurse case managers; APNs: advanced practice nurses; SR: systematic review; LR: bibliographic review; DS: descriptive study; EO: expert opinion; and CA: comparative analysis.

**Table 2 healthcare-12-00749-t002:** Results of chronic disease management strategies by nurses in the different Autonomous Communities of Spain.

Author/Year.	Sample	Main Results	Design	JBI *	JBI **
Ministry of Health, Social Services and Equality. 2012 [35]	CPs	Addressing chronicity requires promoting interdisciplinary teamwork, guaranteeing continuity of care with the maximum participation of the patient and their environment.	CCP ^‡^	5	A
Andalusian Health Service. 2017 [37]	CPC	Integrating care services around individuals’ needs through comprehensive assessment, planned care, and coordination.	CM model ^‡^	5	A
Andalusian Health Service. 2011 [49]	CPC	Promoting self-responsibility, autonomy, and self-care and achieving the greatest possible well-being and quality of life for the person.	Competence development CM ^‡^	5	A
Ministry of Health of Cantabria. 2015 [53]	CPs	Creation of new professional roles for the care of people with chronic illnesses.	CCP ^‡^	5	A
Castilla La Mancha Health Service (Sescam). 2015 [45]	CPs	Strengthening of primary care in the coordination of care and to reinforce the role of nursing in care of chronic patients.	CCP ^‡^	5	A
**Consejería de Sanidad de Salud de Castilla y León. 2013 [43]**	CPs	A model of integrated patient-centred care and continuity of care is proposed, ensuring quality of care and efficiency.	CCP ^‡^	5	A
Ministry of Health. C. Valenciana. 2014 [39]	CPs	The incorporation of NCMs is proposed as new professional profiles for the integrated care of CCPs and their carers.	CCP ^‡^	5	A
**Servei de Salut Illes Balears.** 2017 [50]	CPs	The creation and strengthening of the figure of the nurse case manager.	CCP ^‡^	5	A
Canary Islands Health Service. 2015 [47]	CPs	Strengthening the competencies of healthcare professionals, applying CM for the most complex patients.	CCP ^‡^	5	A
Riojan Health Service. 2014 [46]	CPs	Consolidation of the model of shared care between PC and hospital care and strengthening case management in the care of high-risk chronic patients with the consequent creation of new professional roles.	CCP ^‡^	5	A
**Consejería de Sanidad de Madrid.** 2013 [51]	CPs	Complex cases should be dealt with by a multidisciplinary team, guaranteeing continuity of care. Strengthening of new nursing roles.	CCP ^‡^	5	A
Murcian Health Service. 2013 [48]	CPs	To promote the role of nurses in chronicity with the implementation of advanced competencies and the development of new professional roles.	CCP ^‡^	5	A
Navarra Department of Health. 2013 [52]	CPs	Creation of new professional profiles oriented to the care of complex chronic patients: case manager and hospital liaison nurse together with telephone counseling nurses.	CCP ^‡^	5	A
Basque Health Service. 2010 [38]	CPs	Continuity of care ensured through multidisciplinary, coordinated, and integrated care between different services, levels of care, and sectors with different approaches to clinical integration.	CCP ^‡^	5	A
Galician Health Service. 2018 [44]	CPs	Two innovations in care: the figure of the professional or team of reference for the chronic patient, and the individualized care plan as the cornerstones of a new organisation of care for chronically ill people.	CCP ^‡^	5	A

* level of evidence; ** grade of recommendation; JBI: Joanna Briggs Institute; CPs: chronic patients; CPC: chronic patients and their carers; NCM: nurse case manager; CM: Case Managers; ^‡^ strategies for approaching chronicity in the different autonomous communities in Spain; and CCP: chronicity care program.

## Data Availability

All necessary data are supplied and available in the manuscript; however, the corresponding author will provide the dataset upon request.

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
