# Peer review of "Models Used by Nurse Case Managers in Different Autonomous Communities in Spain: A Scoping Review"

_healthcare, 2024, doi:10.3390/healthcare12070749_

Round 1

Reviewer 1 Report

Comments and Suggestions for Authors

1. The abstract is well drafted and contains the required information

2 It would be useful to indicate - with an awareness of the ambiguity and complexity of approaches - the basic understanding of Case Management.

3. The abbreviation NCM or CMN appears inconsistently in the work to Nurse Case Manager, this should be sorted out

4. Materials and Methods - the procedure for conducting the study is clearly and comprehensively described

5. The results presented in Table 1, Table 2 and Figure 2 are clear, but it would also be good to briefly describe them in the text, directly below these tables and figure.

6. The discussion is generally well drafted, but it would be useful to add an introductory sentence at the beginning of this section to remind the purpose of the analyses.

Author Response

Dear Reviewer,

We wish to thank you for your constructive comments in this review. Your comments have provided valuable insights to help us refine the content and its analysis.  We try to address the issues raised as best as possible in the document attached.

Best regards,

The authors.

Reviewer 2 Report

Comments and Suggestions for Authors

Dear authors,

It was a pleasure for me to examine this work, which presents a review of the current situation of nurse case managers in the different autonomous communities of Spain. I found it to be a very interesting topic.

With the sole objective of improving the quality of this manuscript, I will allow myself to make some comments:

In the question in section 2.2: “What is the current status of the nurse case manager in different communities?”, I would add “in relation to their professional role”, which is ultimately what we want to define, even homogenize.

In tables 1 and 2, it is advisable to put at the end of each one the meaning of the acronym “JBI”, not just the meaning of * and ** (although it is already indicated in section 2.5)

In the discussion section, the first two paragraphs, as written, are more suited to being included in the results section and part of the text could even be adapted to the introduction; but not to the discussion. Therefore, it is relevant that the results of this review be contrasted more broadly with the corresponding bibliography (in addition to what has already been discussed in the submitted manuscript).

In the strengths and limitations of the study, the limitations are specified, but what those strengths are are not indicated; The strengths mentioned should be detailed.

All the best.

Author Response

(The authors gave the same response as above.)

Reviewer 3 Report

Comments and Suggestions for Authors

Dear authors,

Congratulations on your article. 

The study is pertinent and relevant to the discipline of nursing and to understanding the panorama of case management by nurses.

The article is well-written overall, but needs some rewording. I leave suggestions for improvement for further analysis:

Title: unclear... what is meant by current situation? It should be aligned with the objective and research question. It is suggested (validate the correct translation into English): 

Models Used by Nurse Case Managers in Different Autonomous Communities in Spain: A Scoping Review

Abstract: A framing/introduction sentence is missing from the beginning (before the objective).

The objective presented in the abstract must be aligned with the objective presented at the end of the introduction. And it's not! Write it the same way and clearly! Being a scoping review, the objective presented doesn't seem right... It can't be analysed! It should be in line with the research question and change the action verb. Something like (they can personalise it, of course. The correct English translation must be validated): 

Map the evidence on the characteristics of models used by nurse case managers in the different Spanish Autonomous Communities.

Keywords: you can add a keyword that gives greater visibility to nursing care. Example: Nurses or Nursing Care - as long as it is mostly Mesh-Decs (4 to 6 keywords).

Introduction: 

84: the objective should be revised as explained above (in the abstract)

Method: Lapses, but reproducible.

101-102: Reflect on the research question to include the scoping review mnemonic and make it clearer! Suggested example (validate the correct translation into English): 

What are the characteristics of the models used by nurse case managers in the different Spanish Autonomous Communities?

118-122: a SR should not have a time limitation... They should make a sentence that justifies the time cut-off at 10 years ago... given the development or theoretical framework of the implementation of case management in Spain, for example.

125: documents incorporated into the systematic review? Isn't this a Scoping review (SR)?

129-130: Sentence "... except for certain chronicity management strategies that, due to the absence of recent updates, were included if published prior to 2013." Not clear ? should be clarified !

Results:

145: refer to magemente or competency... (correct syntax and sentence reading).

149-150: the name of the figure doesn't have to be so long (it doesn't have to be the title of the article)

163 - 172: there should be no table immediately after the figure. The content of figure 1 should be presented correctly and then figure 1 should be added. Then correctly present the contents of table 2 and then place table 1.

166: Figure 2?

167: Figure 2?

168-169: include the word "in nursing" or "by nurses" (to make the title more complete)

Table 1: revise the way in which the authors are cited - they shouldn't include the authors' first names (only their surname and year). Revise them all.

Table 1: author E.D. is a journal. Check the best way to cite, as there should be an author who signs the opinion and says that it is In Revista xpto. As a last resort, put the name of the journal in full (do the same in the list of bibliographical references).

170-172: check that all types of study are included in the inclusion criteria, in the methodology.

174-183: Table 2 could be better presented, more descriptive about the results it shows.

183: Is there a figure 2?

184-185: Review the name of the table and the use of the expression "strategies for tackling chronicity" throughout the text. There are clearer and more appropriate options, such as: "strategies for dealing with chronic patients by nurses" or "chronic disease management strategies by nurses".

Discussion: Overall, good.

229-230: the sentence is not clear. How can it not have additional costs? 

Conclusions: on the whole, good.

References:

412: the link in this reference does not lead to a concrete document. It should be revised. Not least because it seems important to localise the relevance of the research having been limited to the last 10 years.

Check that all the references in the list are cited.

Check throughout the text that the citations correspond correctly to the references.

Best wishes for the future!

Author Response

(The authors gave the same response as above.)

Reviewer 4 Report

Comments and Suggestions for Authors

Dear authors, thank you for let me reviewing the manuscript titled

“Current situation of the Nurse Case Manager in different Autonomous Communities of Spain: A Scoping Review”.

Major considerations

From my point of view the manuscript offers an interesting information to Spanish Nurses, but has the flaw that it is limited to an specific country. Acordingly to that it is suggested to re write, and  structured the discusión (with epigraphs), with greater depth, expanding upon new ideas in international aspects. For example, from the national situation and then arguing within the international scope, exploring other topics as the relation between the “nurse specializations” and the “nurse case manager” in Europe and/or other countries. Questions like if they are equivalents or if it is necessary to be firstly a specialist nurse, could be argued. In fact autors mention in line 48 that, “..(ICN) [9] as: "a specialist nurse who has acquired the expert knowledge base..”. In this sense it is suggested to read the article with DOI: 10.1111/inr.12204

Please, authors have to review (one by one) and write correctly all the references. For example, the link in the reference 23 (line 333) is not approppriate:

https://joannabriggs.org/about_jbi/our_approach

This is crucial because could denote negligence.

On other hand, in methodology, in lines 111-116 authors mention Spanish words but do not appear in the search strings (lines 115-116)

In the “Results” section, the epigraphs 3.1. Nurse Case Manager, and 3.2. Implementing a new role: Nurse Case Manager, looks like similars.

Minor considerations.

Please, define in Tables 1 and 2 the acronyms: CGE, CME, QA, CGAs.

Line 108, to delete “18” after “website”.

Line 208, to delete “15” after “date”

Author Response

(The authors gave the same response as above.)

Round 2

Reviewer 4 Report

Comments and Suggestions for Authors

Congratulations to authors. The modifications have improved the manuscript.